# Lyme Neuroborreliosis—Significant Local Variations in Incidence within a Highly Endemic Region in Sweden

**DOI:** 10.3390/microorganisms11040917

**Published:** 2023-04-01

**Authors:** Per-Olof Nilsson, Ivar Tjernberg

**Affiliations:** 1Medical Programme, Faculty of Medicine and Health Sciences, Linköping University, 58 183 Linköping, Sweden; 2Department of Clinical Chemistry and Transfusion Medicine, Region Kalmar County, and Department of Biomedical and Clinical Sciences, Division of Inflammation and Infection, Linköping University, 581 83 Linköping, Sweden

**Keywords:** Lyme neuroborreliosis, epidemiology, incidence, geographic information system

## Abstract

The aim of this study was to perform a detailed epidemiological overview of Lyme neuroborreliosis (LNB) 2008–2021 in a highly Lyme borreliosis-endemic area in Sweden using a geographic information system (GIS). Diagnosis of LNB was based on clinical symptoms and analysis of cerebrospinal fluid (CSF) according to European guidelines. From laboratory databases and medical records, we detected all patients with CSF pleocytosis and intrathecal anti-*Borrelia* antibody production and listed clinical features. The distribution of LNB cases within Kalmar County, Sweden was investigated using GIS. In total, 272 cases of definite LNB were confirmed with an average yearly incidence of 7.8/100,000. Significant differences in incidence were noted between children 0–17 years (16/100,000) and adults 18+ years (5.8/100,000) (*p* < 0.001), between rural (16/100,000) and urban areas (5.8/100,000) (*p* < 0.001) and between selected municipalities (*p* < 0.001). Distinct clinical differences in presentation of LNB were also noted between children and adults. Thus, the incidence of LNB varies significantly locally and in relation to age, and clinical presentation shows differences between children and adults. Surveillance of LNB and knowledge of local epidemiological conditions may facilitate preventive measures.

## 1. Introduction

Lyme borreliosis (LB) is the most common tickborne infection in Europe and North America, and it is anticipated to further increase due to climate change [1,2]. It is caused by the bacteria *Borrelia burgdorferi* sensu lato complex, including *Borrelia afzelii*, *Borrelia garinii* and *Borrelia burgdorferi* sensu stricto, species commonly causing European LB carried by the tick *Ixodes ricinus* (*I. ricinus*) [2]. Examples of additional species that may cause human infection include *Borrelia bavariensis*, *Borrelia bissettii*, *Borrelia lusitaniae*, *Borrelia spielmani* and *Borrelia valaisiana* [3]. Exposure to *Borrelia* may heal spontaneously subclinically or present itself with various symptoms [4]. LB can manifest in different ways: the most common is dermatological erythema migrans and the second most common in Europe when the infection reaches the peripheral and/or the central nervous system is Lyme neuroborreliosis (LNB) [2,5]. Additional clinical manifestations include Lyme arthritis, acrodermatitis chronica atrophicans, borrelial lymphocytoma and Lyme carditis [2,6]. An interesting association between the dermatological acrodermatitis chronica atrophicans and peripheral neuropathy has also been reported [7]; thus, patients may suffer multiple manifestations in parallel. Although LNB is not a notifiable disease in Sweden, it is still considered the most common cause of bacterial central nervous system infection in Sweden [8,9].

The incidence of LB in Sweden has been reported as 69/100,000, varying between regions depending on several factors such as exposure and behaviour in the human population, density of ticks carrying bacteria, animals acting as reservoirs of *Borrelia* and climate factors such as temperature and humidity [8,10]. The highest regional incidence in Sweden has been reported in Kalmar County, with an incidence of 160/100,000 [8].

European LNB may present as meningoradiculitis, showing symptoms and signs of painful radiculitis, facial palsy and headache. Less common clinical features are encephalitis, myelitis and other cranial neuropathies. Interestingly, painful radiculitis is only occasionally found in American patients; thus, clinical characteristics vary between American and European LNB [5,11]. Symptoms last between a few weeks to up to several months, usually healing shortly after an antibiotic treatment with doxycycline or ceftriaxone [12,13,14]. However, LNB may cause long-term issues, as a 5-year follow-up study has shown that 25% of participants experienced residual neurological symptoms five years after treatment of the infection [15].

Diagnosis of LNB is based on clinical symptoms and analysis of cerebrospinal fluid (CSF) according to European Federation of Neurological Societies (EFNS) guidelines. Diagnosis of LNB is definite when the following three criteria are fulfilled; (I) neurological symptoms of LNB with no other cause; (II) CSF pleocytosis; (III) *Borrelia*-specific antibodies produced intrathecally [11]. Other tests such as CXCL13 in CSF may give further support for the diagnosis but have not yet been introduced in diagnostic guidelines and recommendations.

The incidence of LB and LNB is tightly tied to the population of *I. ricinus* and the frequency of ticks carrying *Borrelia*. From earlier studies of Kalmar County, we know that climate is an important factor, with mild winters and a longer vegetation period of +5 °C being significant for the active season for *I. ricinus* [16,17]. The risk of tick bites correlates with time spent outdoors and the highest amount of reported tick bites correlates with the summer period of June–August in south-eastern Sweden [18]. The risk of contracting LB from a tick bite increases with the time the tick has been feeding, with detection and removal of ticks being above 12 h in a majority of cases and more than 24 h in a third of the cases [19].

The specific species distribution of *Borrelia* can vary between regions depending on the availability and type of reservoir host. Although all relevant *Borrelia* species may cause all clinical manifestations, the different genospecies express various levels of organotropism. During its life cycle from larva to nymph to adult, *I. ricinus* may feed on most vertebrates, and depends on small-to-medium-sized wildlife as carriers and reservoirs of *Borrelia* between larva and nymph stages in their life cycle [20]. Of the different species, *Borrelia garinii* is the one most associated with LNB in Europe [2]. Birds are known as competent carriers of *Borrelia garinii*, and areas with rich bird populations might therefore increase the risk of LNB, whereas Borrelia afzelii is more associated with the reservoirs mice and voles and clinically Borrelia afzelii is more commonly associated with erythema migrans [2,21,22].

The geographic information system (GIS) is a powerful computer system tool to visualise various forms of data in relation to geographic information in an easy-to-understand form. Presenting information using GIS provides a viewpoint that makes it possible to detect connections that otherwise would be difficult to find only using tables to present information.

Considering that Kalmar County is a highly endemic area for LB and LNB, the gathered knowledge about local variations within the County is limited [23]. Moreover, reports comparing paediatric with adult LNB are scarce. Thus, the aim of this study was to perform a detailed epidemiological overview of clinical characteristics in children and adults with LNB in Kalmar County Sweden, and to investigate possible local differences in incidence using GIS.

## 2. Materials and Methods

Cases of LNB were identified as follows: Patients with laboratory results compatible with LNB were identified using the laboratory databases from the Departments of Clinical Microbiology and Clinical Chemistry and Transfusion Medicine at Kalmar County Hospital. Patients with CSF pleocytosis (defined as total CSF white blood cell count > 5 × 10^6^/L) and a positive IgM and/or IgG *Borrelia*-specific CSF/serum antibody index (IDEIA Lyme neuroborreliosis Kit, ThermoFisher Scientific, Waltham, MA, USA) were identified. The IDEIA kit identifies IgM and IgG antibodies to purified native flagellum as antigen [24]. Clinical characteristics for these cases were then systematically collected using information from the electronical medical records. Data sets covered the years 2008–2021 and were chosen due to consistent laboratory routines for diagnosis of LNB for the period. Thus, all cases were confirmed as definite LNB with clinical diagnosis and a positive test for pleocytosis and serology for intrathecal *Borrelia* antibodies according to EFNS guidelines [11]. Medical record data were limited to one month before test date and one year after test date and covered hospital care in Kalmar County (Kalmar, Oskarshamn and Västervik hospitals), and all medical record reviews were performed by Per-Olof Nilsson. We also tracked all occasions of follow-up treatment after each confirmed case. Follow-up treatment was considered each time there was a planned check-up after initiation of treatment with antibiotics, e.g., an evaluation of facial palsy progress. Instances in which a patient contacted hospital care for lingering symptoms that resulted in an in-hospital evaluation were also considered follow-up treatment.

For GIS analysis sampling, date and identity were used to extract patient location by comparing the test date with census registers. Population data for municipalities were gathered from Statistics Sweden (Statistikmyndigheten, SCB) and referred to the population at the end of 2021. Cases were also divided according to residential address either in urban or rural areas according to the definition used by SCB. An urban area was defined as an area with continuous buildings with no more than 200 m between houses and at least 200 residents, and a nonurban area was defined as rural [25]. GIS maps were created using ArcGIS Desktop/Arcmap 10.7.1.

### 2.1. Exclusions

Sampling data not supported by clinical diagnosis and/or by both positive CSF pleocytosis and a positive anti-*Borrelia* antibody index were removed from the database. This included when the same patient had his/her CSF tested more than once, e.g., when there were two or more positive tests for pleocytosis and antibody index, but the clinical diagnosis showed that no new infection had incurred between first and second sampling date. This patient was then only counted once. Cases where symptoms and causes of CSF pleocytosis and anti-*Borrelia* antibody index were deemed as being other than an ongoing LNB were also excluded, e.g., HSV infection or lingering symptoms from an earlier LNB infection. All cases not living in Kalmar County according to population registers were removed in both data gathered from medical charts and for GIS-analysis.

### 2.2. Ethics

This study was ethically approved by the Swedish ethical review authority DNR: 2021-04405.

### 2.3. Statistics

Statistica version 13.5.0.17 was used for statistical calculations with chi-square when comparing frequencies between groups. SPSS version 28.0.0.0 was used for Mann–Whitney U-test when comparing duration of illness before lumbar puncture (LP) in children (0–17 years) and adults (18+ years). A *p*-value < 0.05 was accepted as a significant result.

## 3. Results

Data from the Departments of Clinical Microbiology and Clinical Chemistry and Transfusion Medicine at Kalmar County hospital with a positive CSF to serum anti-*Borrelia* antibody index (IgM and/or IgG) in combination with CSF pleocytosis were gathered and crosschecked, resulting in 302 cases with positive results during the period 2008–2021. This list was checked against medical records, and all cases where positive tests could be caused by other causes than LNB or that did not represent a unique case were excluded, *n* = 9, leaving 293 cases. Cases not registered as residents in Kalmar County during the time of testing were also excluded, *n* = 21, leaving a total of 272 unique remaining definite LNB cases.

Cases showed an annual variation during the study period, with the highest number of confirmed cases in 2016 at 30 cases and the lowest number of cases in 2012 and 2018 with 13 cases (Figure 1a). When divided by age in increments of 10 years, the largest group was children between ages 0–9, representing 33% of the total cases; and the age group of 60–69 was the second largest, representing 20% of the total cases in a bimodal manner (Figure 1b). The number of LNB cases according to month of diagnosis shows a rise in June and has its peak in August, and then falls back during autumn with the fewest number of cases in January-March (Figure 1c).

Confirmed cases were divided and tested for relationship by sex (male, female) and age (children 0–17, adults 18+ years). Chi-square testing showed no significant difference in sex distribution between age groups (*p* = 0.205). Duration of symptoms before LP was longer in the adult group, with a median time before LP of 16 days in adults compared with 7 days among children (*p* < 0.001). It was more common with follow-up contact after treatment in children, while it was more common for the adult group to have residing symptoms after finishing treatment and follow-up (*p* < 0.001) (Table 1).

While adults in relation to children more often presented with radiculitis (*p* < 0.001) and muscle/joint pain (*p* = 0.006), the opposite was found regarding fever (*p* < 0.001), fatigue (*p* < 0.001) and headache (*p* < 0.001), which were more common among children. No significant differences between adults and children were detected for the occurrence of facial palsy, other cranial nerve palsy, meningitis, nausea or erythema migrans (Table 1).

The overall annual incidence in Kalmar County was calculated to 7.8/100,000, with the highest incidence in the municipality of Mörbylånga at 15/100,000, and the lowest in Högsby, Kalmar and Mönsterås municipalities at 4.8–5.1/100,000 (*p* < 0.001). Place of residence in rural or urban areas also showed a difference, as the incidence was 5.8/100,000 in urban areas, whereas it was 16/100,000 in rural areas (*p* < 0.001). A significant difference was detected comparing LNB incidence among children and adults, with 5.8/100,000 in adults and 16/100,000 in children (*p* < 0.001) (Table 2).

### Geographical Analysis

Confirmed cases of LNB were plotted into GIS maps according to the residential address established by population registers at the respective time of diagnosis. Four maps are presented here (Figure 2, Figure 3, Figure 4 and Figure 5). With a topographical underlay, it is possible to visualise the positioning of cases in reference to the surrounding environment. Most cases were clustered in urban areas without other obvious associations to topography (Figure 2).

When coded according to urban or rural area, we detected 162 cases in urban areas and 110 cases living in rural areas. Borgholm and Torsås showed a high number of cases in rural areas, while most cases in Emmaboda and Oskarshamn were clustered in urban areas (Figure 3).

We also chose to group cases by comparing children (0–17) and adults (18+) (Figure 4). Here, adult cases cluster in the Torsås municipality, with few cases in children noted, which is further confirmed in the numbers shown in Table 3. For the remaining municipalities, paediatric cases were in dominance.

In Figure 5, the mean annual incidence for each municipality is visualised in a colour scale enabling an easy overview of the differences in incidences across municipalities and how they are situated geographically in relation to each other. The three municipalities with the highest incidence (Emmaboda, Torsås and Mörbylånga) cluster in the southern part of Kalmar County, surrounding the neighbouring Kalmar municipality, which has one of the lowest incidence rates. The relationships between urban/rural place of residence were approximately the same across municipalities with generally higher incidence rates in rural areas, except for Emmaboda, which had more similar incidence rates between rural and urban areas (Table 3).

## 4. Discussion

In this study, we investigated the regional epidemiology of LNB in the highly endemic Kalmar County, Sweden during the last 14 years (2008–2021). In addition to demographical and clinical parameters, we have also mapped the cases according to address of residence at the time of diagnosis with interesting findings. In our study, we detected an overall incidence of 7.8/100,000 of LNB within Kalmar County. This is higher than the national average in Sweden at 6.3/100,000, and more than double that of the neighbouring country, Denmark, at 3.2/100,000 [26,27].

The main clinical findings include a difference in the presentation of LNB between children and adults. While radiculitis and muscle/joint pain were more common in adults, fever, fatigue and headache were reported more commonly in children. This is in line with earlier findings in Jönköping County [28]. In addition, persistent symptoms from LNB were more often detected at follow-up in adults, suggesting a slower or reduced rate of recovery compared with children. This could possibly also be a result of the difference in time of diagnosis and antibiotic treatment initiation in relation to symptom duration, in which adults generally have longer symptom duration by diagnosis compared with children. An association between LNB symptom duration by start of antibiotic treatment and rate of sequelae has been reported previously [15].

Sex was not found to influence LNB incidence in this material. However, the incidence of LNB in children, 16/100,000 was more than double that of adults, 5.8/100,000, which is important information considering the possibility of targeted pre-emptive actions. We were also able to confirm the highest-risk groups for LNB being in the age ranges of 0–9 years with the highest incidence, followed by a second spike at 60–69 years [16,28].

Furthermore, this study also shows that there is a seasonality of LNB, with a clear majority of cases during the autumn, with the peak number of cases in August. However, cases may be diagnosed all year, even during the winter months; thus, the diagnosis cannot be ruled out any period of the year.

The number of cases in each municipality shows that clear differences and population density does not seem to be the most important determinant. Kalmar municipality, the most densely populated within the County, compared with Mörbylånga municipality, which has a three times higher incidence rate, shows that other factors are in play. Urban or rural place of residence seems to be an important factor, as rural areas represented 40% of all LNB cases (*n* = 110/272) while only representing 20% of the total population in the County. Living closer to nature seems to be an important factor, as a Danish study on the island of Funen tied a higher risk for LNB to closeness to forested areas [29]. However, the generally increased risk for LNB associated with rural area of residence may obviously be counteracted or balanced by other factors, as Mörbylånga, with 78.4 % of its population in urban areas, still showed a higher incidence than Borgholm, with 51.7 % of its population in urban areas, the lowest in Kalmar County. Furthermore, the municipality of Högsby, the third lowest in urban population percentage (64.6%), has the same incidence of LNB as the municipality of Kalmar, the municipality with the highest percentage of urban population (89.3%).

One possible explanation for this discrepancy is the large number of migratory birds on southern Öland and in extension Mörbylånga municipality and their role as a vector for spreading tick-borne disease, and that birds are more suited as carriers for *B. garinii,* which is the genospecies of *Borrelia* most likely to cause LNB [2,21,30]. The most commonly found *Borrelia* species in ticks in Sweden has been reported to be *Borrelia afzelii* (61%), followed by *Borrelia garinii* (23%) [31]; however, it is possible that local and regional distributions of *Borrelia* species may vary, as indicated by our present findings. We do know that birds found at Ottenby Bird Observatory, which is situated within Mörbylånga municipality, are competent carriers of *B. garinii* [30]. Other municipalities might have a different type of main reservoir host carrying a genospecies of *Borrelia* less likely to cause LNB. The tick distribution and density in different areas should also be considered, which will affect the risk of tick bites, and in extension, cases of LNB. A known influence on tick population is the presence and movement of deer, which needs to be considered when evaluating the number of ticks in different municipalities [32]. Discrepancies between groups of urban populations may also be explained by our definition used, as the max range of 200 m to nearest neighbour may look very different from the least to the most densely populated areas.

In addition to the above-mentioned parameters, behavioural differences also need to be taken in consideration, as there might be regional differences in tick exposure as well as the tendency to seek medical advice in case of LB-associated symptoms and signs. In addition, potential differences may also exist in medical routines and traditions across different medical clinics.

Obviously, the risk of LNB is multifactorial, as well as the reasons for detected differences in this study. Here we show that factors such as age, geography and urban or rural place of residence influence the risk of LNB. Human exposure to tick bites is also, of course, a natural risk factor, although this study was not designed to compare potential differences in this aspect across various geographical areas. Further studies are needed to find the reasons for the large variations in local incidence.

To enable future epidemiological studies both within Sweden but also across Europe, we would benefit from a uniform and comprehensive adherence to the surveillance system of LNB by the European Centre for Disease Prevention and Control [33]. In the absence of a complete surveillance system according to the EFNS criteria, a laboratory notification system of LNB based on the presence of intrathecal anti-*Borrelia* antibodies would be valuable and reasonably easy to implement [26].

A main strength of this study is that we have been able to work with a large number of definite LNB patient cases. However, as strict diagnostic criteria were applied in the study, additional cases may exist in whom lumbar puncture has not been performed for definite diagnosis, e.g., suspected LNB cases in primary health care. The extent, characteristics and distribution of such cases are obviously difficult to determine, but may to a limited extent effect the findings in case of an uneven distribution across the County. For the GIS analysis, we have used the residential address at time of diagnosis as a proxy for where the tick bite might have occurred. Naturally, this estimation, and our findings with regard to geography, must be interpreted with caution.

## 5. Conclusions

The distribution of clinical signs and symptoms in the studied LNB cases confirm significant clinical differences between children and adults. Epidemiologically, interesting and large differences in incidence between municipalities in Kalmar County were noted. In addition, place of residence in rural areas seems to be associated with a higher incidence when compared to urban areas. Other factors seem to be in play, such as variations in geography, tick abundance, human behaviour, diagnostic routines, and availability of different reservoir hosts and the local distribution of various *Borrelia* genospecies.

In conclusion, our findings underscore the importance and value of LNB surveillance data for epidemiological studies, which may facilitate targeted preventive actions against LNB.

## Figures and Tables

**Figure 1 microorganisms-11-00917-f001:**
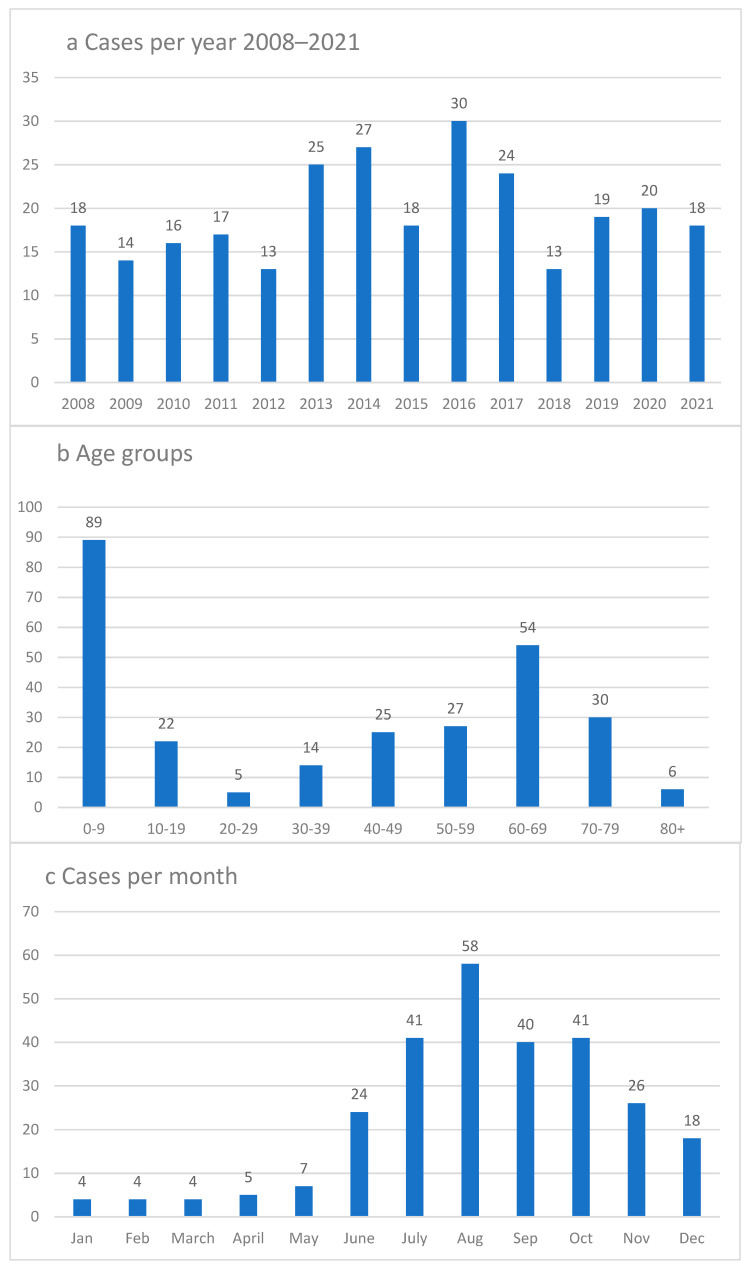
(**a**–**c**). Lyme neuroborreliosis distribution 2008–2021, number of cases. (**a**): Cases annually 2008–2021. (**b**): Age distribution 2008–2021. (**c**): Case distribution by month 2008–2021.

**Figure 2 microorganisms-11-00917-f002:**
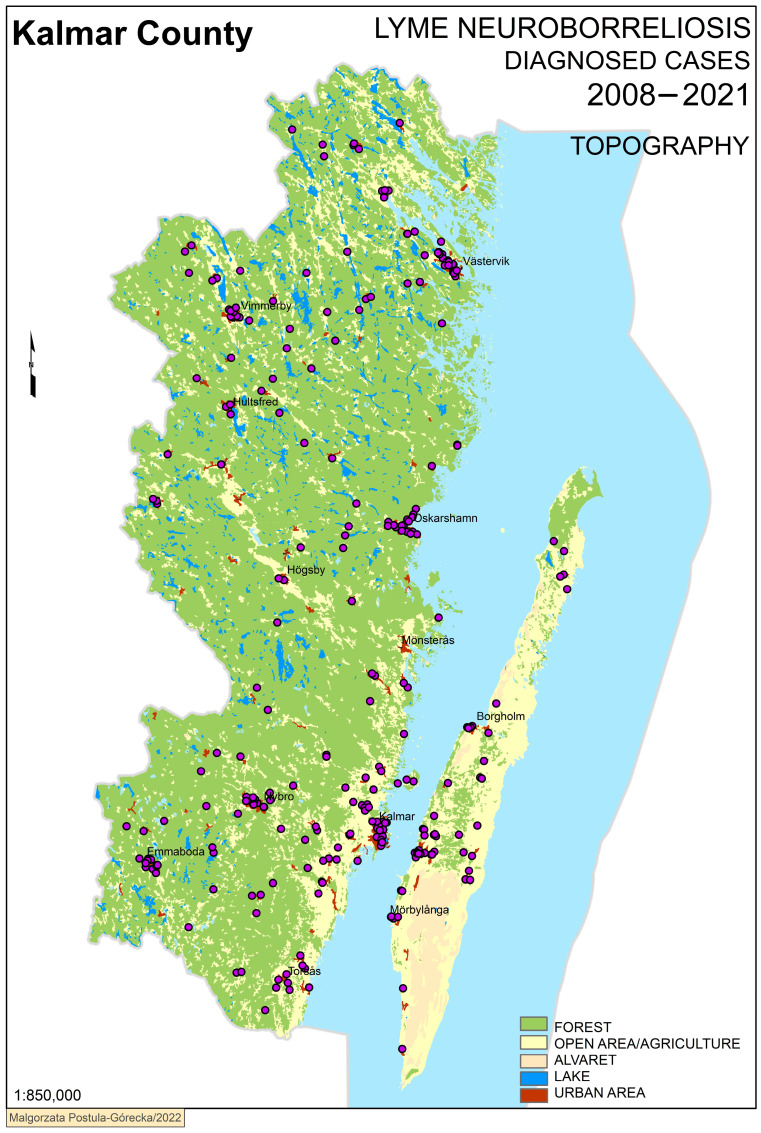
Lyme neuroborrelisos case distribution with topographical overlay within Kalmar County, Sweden.

**Figure 3 microorganisms-11-00917-f003:**
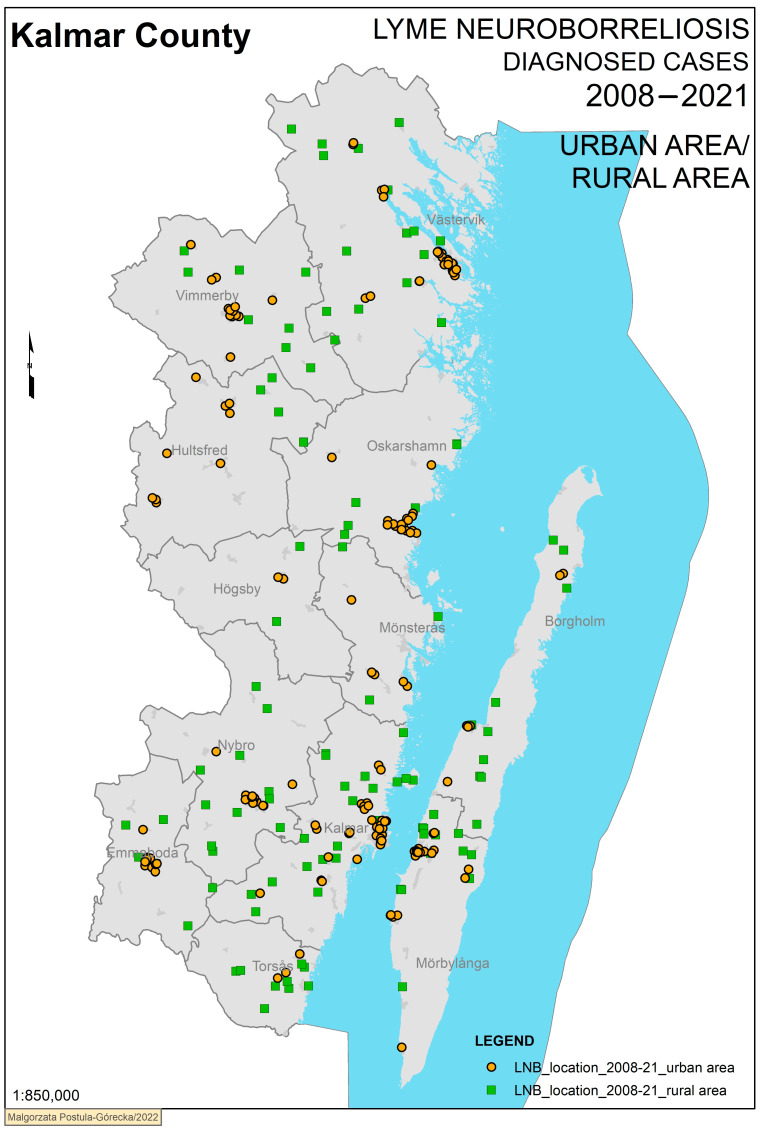
Confirmed Lyme neuroborreliosis cases grouped according to urban or rural area status. Figure 3. Definite Lyme neuroborreliosis cases in Kalmar County 2008–2021 grouped by living in urban or rural areas, with 162 cases in urban areas and 110 in rural areas. The municipalities within Kalmar County are shown in grey.

**Figure 4 microorganisms-11-00917-f004:**
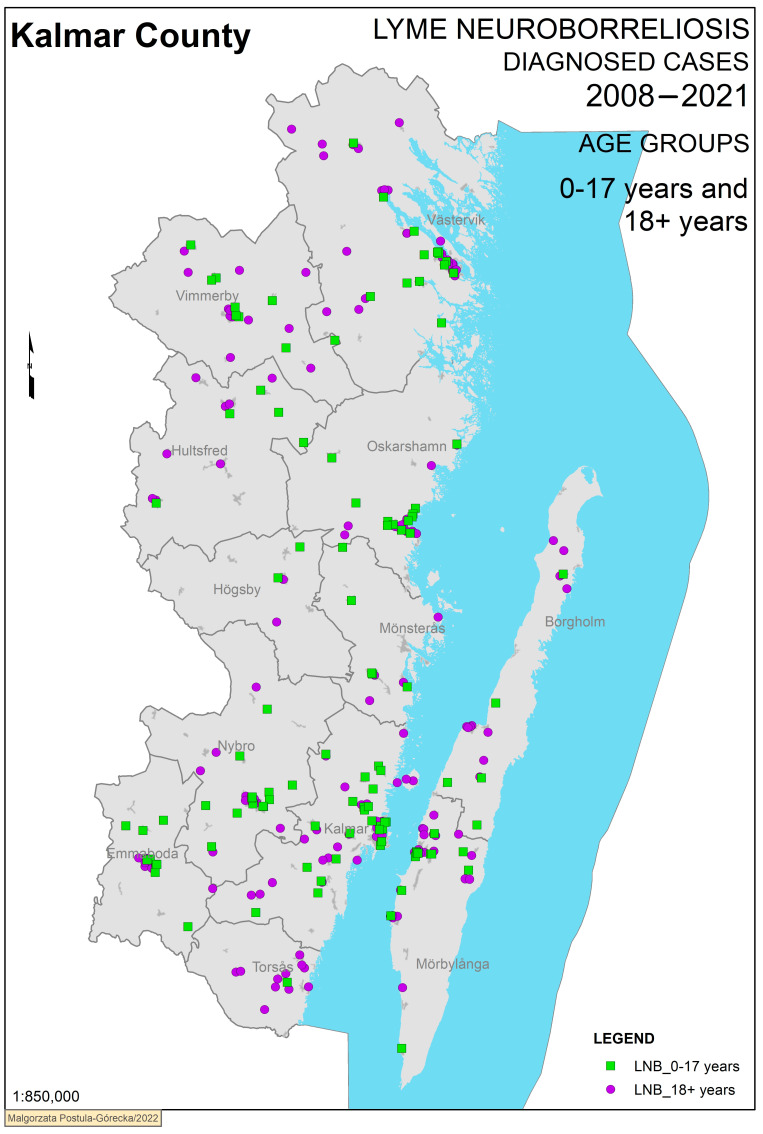
Definite cases of Lyme neuroborreliosis grouped by children and adults. Figure 4. Definite Lyme neuroborreliosis cases in Kalmar County 2008–2021 grouped by children (0–17 years) and adults (18+ years). The municipalities within Kalmar County are shown in grey.

**Figure 5 microorganisms-11-00917-f005:**
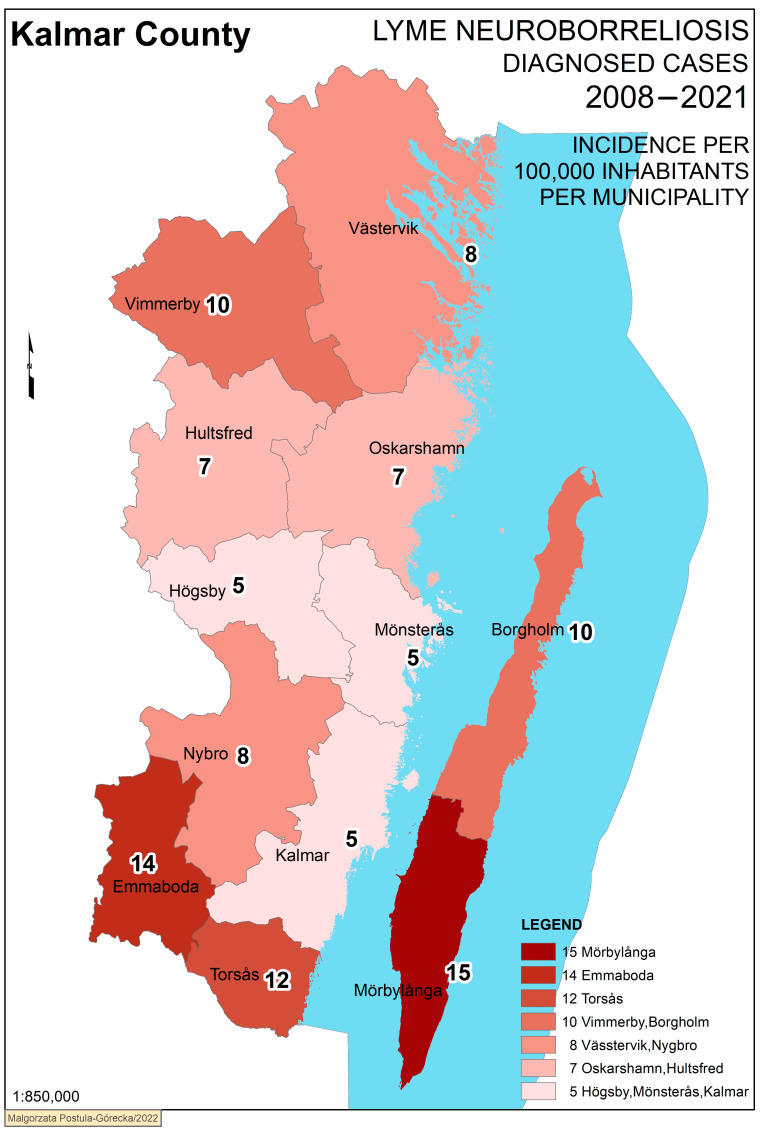
Mean annual incidence in each municipality.

**Table 1 microorganisms-11-00917-t001:** Definite LNB cases divided by age and distribution of symptoms.

Patient Group and Symptoms			
Age Group, Years	Children, 0–17 (*n* = 111)	Adults, 18+ (*n* = 161)	*p*-Value
Median age (range), years	6 (0–17)	61 (20–91)	
Female/male, *n* (% female)	59/52 (53%)	73/88 (46%)	0.205
Duration of symptoms before	7 (1–90)	16 (1–150)	<0.001
treatment days, median (range)			
Facial palsy, *n* (%)	63 (57%)	78 (48%)	0.178
Other cranial nerve palsy, *n* (%)	3 (3%)	10 (6%)	0.183
Radiculitis, *n* (%)	5 (5%)	77 (48%)	<0.001
Meningitis, *n* (%)	25 (23%)	25 (16%)	0.143
Fever, *n* (%)	44 (40%)	9 (6%)	<0.001
Fatigue, *n* (%)	61 (55%)	40 (25%)	<0.001
Headache, *n* (%)	58 (52%)	42 (26%)	<0.001
Muscle/joint pain, *n* (%)	36 (32%)	79 (49%)	0.006
Nausea, *n* (%)	18 (16%)	14 (9%)	0.059
Erythema migrans, *n* (%)	19 (17%)	22 (14%)	0.434
Follow-up of treatment, *n* (%)	97 (87%)	113 (70%)	<0.001
Symptoms at follow-up, *n* (%)	9 (8%)	44 (27%)	<0.001

Statistical analysis was performed using chi-square testing except for duration of symptoms before treatment, for which Mann–Whitney U-test was used. LNB, Lyme neuroborreliosis.

**Table 2 microorganisms-11-00917-t002:** Comparison of average LNB annual incidence for the period of 2008–2021.

			Incidence Annually per		Population	Mean
	Cases	Population	100,000 Inhabitants	*p*-Value	Urban/Rural (%)	Age (Years)
**Kalmar County**	272	247,175	7.8		80.1/19.9	44.4
Sex				0.741		
Male	140	124,754	8.0			
Female	132	122,421	7.7			
**Age group**				<0.001		
Children (0–17)	111	48,879	16			
Adults (18+)	161	198,296	5.8			
**Urban/Rural**				<0.001		
Urban	162	197,906	5.8			
Rural	110	49,263	16			
**Municipality**				<0.001		
Borgholm	16	10,895	10		51.7/48.3	51.4
Emmaboda	18	9329	14		69/31	46.1
Hultsfred	13	14,056	6.6		81.8/18.2	45.6
Högsby	4	5645	5.1		64.6/35.4	45.6
Kalmar	51	71,328	5.1		89.3/10.7	41.1
Mönsterås	9	13,258	4.8		75.8/24.2	45.1
Mörbylånga	34	15,722	15		78.4/21.6	45.2
Nybro	24	20,284	8.4		79/21	44.3
Oskarshamn	26	27,220	6.8		85.3/14.7	43.8
Torsås	12	7113	12		58.7/41.3	46.5
Vimmerby	22	15,578	10		73.2/26.8	44.6
Västervik	43	36,747	8.4		81.1/18.9	46.7

Statistical analysis was performed using chi-square testing. Urban area was defined as an area with continuous buildings with no more than 200 m between houses and at least 200 residents and a nonurban area was defined as rural. LNB, Lyme neuroborreliosis.

**Table 3 microorganisms-11-00917-t003:** Average annual LNB incidence per 100,000 inhabitants, overall and by municipality for the period 2008–2021.

	Total	0–17	18+	Urban	Rural
Kalmar County, overall	7.8	16	5.8	5.8	16
**Municipality**					
Borgholm	10	22	8.4	7.6	14
Emmaboda	14	36	8.5	13	16
Hultsfred	6.6	13	5.1	5.6	11
Högsby	5.0	13	3.2	3.9	7.2
Kalmar	5.1	12	3.4	3.4	20
Mönsterås	4.8	14	2.7	4.3	6.7
Mörbylånga	15	25	13	12	27
Nybro	8.4	22	4.9	5.8	18
Oskarshamn	6.8	16	4.6	6.2	11
Torsås	12	5.3	17	5.1	22
Vimmerby	10	21	7.4	8.1	15
Västervik	8.4	15	6.9	6.5	16

Urban area is defined as an area with continuous buildings with no more than 200 m between houses and at least 200 residents and nonurban area is defined as rural. LNB, Lyme neuroborreliosis.

## Data Availability

Anonymised data presented in this study are available on request from the corresponding author. The data are not publicly available due to conditions of the ethical permission.

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
