# Peer review of "Lyme Neuroborreliosis—Significant Local Variations in Incidence within a Highly Endemic Region in Sweden"

_microorganisms, 2023, doi:10.3390/microorganisms11040917_

Round 1

Reviewer 1 Report

This is a very interesting study that contributes to our expending knowledge about Lyme disease and in particular neuroborreliosis. The authors are to be congratulated on this excellent work. I have the following suggestions for authors in order to improve the paper further:

1. Line 32- LD is the most common tick borne disease in North America as well, and it is anticipated that cases will increase for 20% as a result of climate change-  suggest adding that ( https://pubmed.ncbi.nlm.nih.gov/30473737/)

2. Line 44- reference is missing after statement that climate influence the incidence of the disease

3. Line 48- references are missing after the statement regarding less common clinical features such as acute transverse myelitis , encephalitis etc. Please add appropriate citations here. It also should be added that clinical presentation vary between European and North American patients. For example, painful radiculoneuritis is very rare in North America ( https://pubmed.ncbi.nlm.nih.gov/29383323/)

4. Line 104- expend abbreviation PON 

5. Discussion- level of parental education might be explored briefly as some studies have shown that more educated families are more likely to see medical attention, hence- more likely to be diagnosed. 

Author Response

Dear Sir, 

Thank you very much for your review of this manuscript providing valuable opinions and suggestions in order to improve the quality. Please note our responses point-by-point in red below as well as the revised manuscript: 

This is a very interesting study that contributes to our expending knowledge about Lyme disease and in particular neuroborreliosis. The authors are to be congratulated on this excellent work. I have the following suggestions for authors in order to improve the paper further:

     1.Line 32- LD is the most common tick borne disease in North America as well, and it is anticipated that cases will increase for 20% as a result of climate change-  suggest adding that ( https://pubmed.ncbi.nlm.nih.gov/30473737/)

Thank you for this suggestion, we have added this to the introduction and also included the suggested reference.

  1. Line 44- reference is missing after statement that climate influence the incidence of the disease

Indeed, thank you for pointing this out. This statement has now been supported by reference no 10 in the revised manuscript.

  1. Line 48- references are missing after the statement regarding less common clinical features such as acute transverse myelitis , encephalitis etc. Please add appropriate citations here. It also should be added that clinical presentation vary between European and North American patients. For example, painful radiculoneuritis is very rare in North America ( https://pubmed.ncbi.nlm.nih.gov/29383323/)

This is true and worth to note for the readers, please see the adjusted text in this paragraph with the addition of references.

  1. Line 104- expend abbreviation PON 

This is the abbreviation of one of the co-authors of the manuscript, we have now expanded to his full name.

  1. Discussion- level of parental education might be explored briefly as some studies have shown that more educated families are more likely to see medical attention, hence- more likely to be diagnosed. 

Thank you for this suggestion. However, we feel that this issue regarding differences in patterns of seeking medical advice is already covered in one of the paragraphs in the Discussion (lines 302-306 of the revised manuscript) and that expanding the discussion in this particular issue lies somewhat out of the scope of the article. 

Reviewer 2 Report

The work carries out a survey of the incidence of neuroborreliosis in Kalmar County in Sweden.

There are the following observations:

introduction:

Lines 33-34: Borrelia brurgdorferi sensu lato (Borreliæ Lyme Group) in Europe, are not only those indicated, but also Borrelia valaisianaB. lusitaniaeB. bavarensisB. spielmani and in particular also B. valaisiana can cause the Acrodermatitis chronica atrophicans (Trevisan G, Cinco M, Trevisini S, di Meo N, Chersi K, Ruscio M, Forgione P and Bonin S. Borreliae Part 1: Borrelia Lyme Group and EchidnaReptile Group. Biology 2021, 10, 1036. https ://doi.org/10.3390/biology10101036).

Lines 36-38: The myo-articular ones must be added to the cutaneous and neurological manifestations of Lyme (Trevisan G, Ruscio M, Bonin S. A Practical Approach to the Diagnosis of Lyme Borreliosis: From Clinical Heterogeneity to Laboratory Methods. Frontiers in Medicine. REVIEW published: xx June 2020 doi: 10.3389/fmed.2020.00265).

Line 48. To the neurological manifestations, peripheral ones must also be added, which are infrequent and often derive from Acrodermatitis chronica atrophicans, (Ogrinc K, Maraspin V. Nervous system involvement in Lyme borreliosis. Open Dermatol J. (2016) 10:44– 54. doi: 10.2174/1874372201610010044 - Kindstrand E, Nilsson BY, Hovmark A, Pirskanen R, Åsbrink E. Peripheral neuropathy in acrodermatitis chronica atrophicans - effect of treatment. Acta Neurol Scand. 2002 Nov;106(5):253-7 doi: 10.1034/j.1600-0404.2002.01336.x.), which in Sweden has been extensively described by Eva Åsbrink (Åsbrink E, Hovmark A, Hederstedt B. The spirochetal etiology of acrodermatitis chronica atrophicans Herxheimer. Acta Derm Venereol. 1984;64(6):506-12.)

Materials and Methods:

The antigens tested by the kit used must be specified, in particular for IgG (early antigens such as VlsE, Flagellin (p41 / p14), late DbpA (p17) and very late (p83/100).

Discussion:

Lines 273-276. This is a very interesting fact. It is important to comment on the data in relation to the Borreliæ Lyme Group present in Sweden: Eva Åsbrink indicated an important presence of Borrelia afzelii, the main cause of Acrodermatitis chronica atrophicans, while in the Kalmar area there is an important diffusion of Borrelia garinii, probably as imported by migratory birds. This has already been reported for the Faroe Islands, where migratory birds are infested by ticks, carrying Borrelia garinii (Gylfe, Å; Olsen, B.; Strasevicius, D.; Marti Ras, N.; Weihe, P.; Noppa , L.; Ostberg, Y.; Baranton, G.; Bergstrom, S. Isolation of Lyme disease Borrelia from puffins (Fratercula arctica) and seabird ticks (Ixodes uriae) on the Faeroe Islands. J. Clin. Microbiol. 1999, 37 , 890–896, doi:10.1128/JCM.37.4.890–896.1999.).

Author Response

Dear Sir, 

Thank you very much for your review of this manuscript providing valuable opinions and suggestions in order to improve the quality. Please note our responses point-by-point in red below as well as the revised manuscript: 

The work carries out a survey of the incidence of neuroborreliosis in Kalmar County in Sweden.

There are the following observations:

introduction:

Lines 33-34: Borrelia brurgdorferi sensu lato (Borreliæ Lyme Group) in Europe, are not only those indicated, but also Borrelia valaisianaB. lusitaniaeB. bavarensisB. spielmani and in particular also B. valaisiana can cause the Acrodermatitis chronica atrophicans (Trevisan G, Cinco M, Trevisini S, di Meo N, Chersi K, Ruscio M, Forgione P and Bonin S. Borreliae Part 1: Borrelia Lyme Group and Echidna‐Reptile Group. Biology 2021, 10, 1036. https ://doi.org/10.3390/biology10101036).

Thank you for pointing this out, we are aware of this, and the intention was only to give examples of relevant species. We have now included additional species as requested together with the suggested reference. 

Lines 36-38: The myo-articular ones must be added to the cutaneous and neurological manifestations of Lyme (Trevisan G, Ruscio M, Bonin S. A Practical Approach to the Diagnosis of Lyme Borreliosis: From Clinical Heterogeneity to Laboratory Methods. Frontiers in Medicine. REVIEW published: xx June 2020 doi: 10.3389/fmed.2020.00265).

Thank you for the suggestion. The introduction has been updated including additional clinical manifestations. The suggested reference has also been included.

Line 48. To the neurological manifestations, peripheral ones must also be added, which are infrequent and often derive from Acrodermatitis chronica atrophicans, (Ogrinc K, Maraspin V. Nervous system involvement in Lyme borreliosis. Open Dermatol J. (2016) 10:44– 54. doi: 10.2174/1874372201610010044 - Kindstrand E, Nilsson BY, Hovmark A, Pirskanen R, Åsbrink E. Peripheral neuropathy in acrodermatitis chronica atrophicans - effect of treatment. Acta Neurol Scand. 2002 Nov;106(5):253-7 doi: 10.1034/j.1600-0404.2002.01336.x.), which in Sweden has been extensively described by Eva Åsbrink (Åsbrink E, Hovmark A, Hederstedt B. The spirochetal etiology of acrodermatitis chronica atrophicans Herxheimer. Acta Derm Venereol. 1984;64(6):506-12.)

We appreciate this good suggestion. We have clarified that both the peripheral and/or the central nervous system may be involved in Lyme neuroborreliosis, and that ACA may be associated with peripheral neuropathy. The sentences on lines 40-41 and 44-46 in the revised manuscript have been changed accordingly, and the references by Ogrinc and Maraspin and Kindstrand et al have been inserted.

Materials and Methods:

The antigens tested by the kit used must be specified, in particular for IgG (early antigens such as VlsE, Flagellin (p41 / p14), late DbpA (p17) and very late (p83/100).

The requested information has been added to the manuscript on lines 106-107 together with reference no 25.

Discussion:

Lines 273-276. This is a very interesting fact. It is important to comment on the data in relation to the Borreliæ Lyme Group present in Sweden: Eva Åsbrink indicated an important presence of Borrelia afzelii, the main cause of Acrodermatitis chronica atrophicans, while in the Kalmar area there is an important diffusion of Borrelia garinii, probably as imported by migratory birds. This has already been reported for the Faroe Islands, where migratory birds are infested by ticks, carrying Borrelia garinii (Gylfe, Å; Olsen, B.; Strasevicius, D.; Marti Ras, N.; Weihe, P.; Noppa , L.; Ostberg, Y.; Baranton, G.; Bergstrom, S. Isolation of Lyme disease Borrelia from puffins (Fratercula arctica) and seabird ticks (Ixodes uriae) on the Faeroe Islands. J. Clin. Microbiol. 1999, 37 , 890–896, doi:10.1128/JCM.37.4.890–896.1999.).

Thank you for this comment, this is indeed interesting. We have included a comment on what is known regarding the distribution of different Borrelia species in Swedish ticks from Wilhelmsson et al 2010 in relation to our present findings. See lines 288-291 in the revised manuscript.